# The Proliferation of Chang Liver Cells After Simulated Microgravity Induction

**DOI:** 10.3390/cimb47030164

**Published:** 2025-02-27

**Authors:** Huy Nghia Quang Hoang, Chi Nguyen Quynh Ho, Loan Thi Tung Dang, Nhan Lu Chinh Phan, Chung Chinh Doan, Han Thai Minh Nguyen, Cuong Phan Minh Le, Son Nghia Hoang, Long Thanh Le

**Affiliations:** 1Animal Biotechnology Department, Institute of Tropical Biology, Vietnam Academy of Science and Technology, Ho Chi Minh 700000, Vietnam; hoangnghiaquanghuy@gmail.com (H.N.Q.H.); quynhchihonguyen@gmail.com (C.N.Q.H.); plcnhan@gmail.com (N.L.C.P.); doanchinhchung@gmail.com (C.C.D.); han.nguyen@unh.edu (H.T.M.N.); lephanminhcuong@gmail.com (C.P.M.L.); hoangnghiason@yahoo.com (S.N.H.); 2Biotechnology Department, Graduate University of Science and Technology, Vietnam Academy of Science and Technology, Ha Noi 100000, Vietnam; 3Faculty of Biology and Biotechnology, University of Science, Vietnam National University, Ho Chi Minh 700000, Vietnam; dttloan@hcmus.edu.vn; 4Center for Biotechnology and Genomic Medicine, Medical College of Georgia, Augusta University, Augusta, GA 30912, USA; 5Life Science Department, University of New Hampshire at Manchester, Manchester, NH 03101, USA

**Keywords:** Chang liver cells, simulated microgravity, recovery after simulated microgravity induction, proliferation, cell cycle, viability

## Abstract

This study aimed to assess the recovery capability of Chang liver cells (CCL-13) following simulated microgravity (SMG) induction. CCL-13 cells were cultured under SMG conditions for 72 h, and control group cells were cultured under 1G conditions for an identical duration. Cells from the SMG and control groups were further cultured under 1G conditions and assessed after 24 h and 72 h intervals in the gravity recovery experiment. The WST1 results indicated that CCL-13 proliferation was more evident in the control group than in the SMG group after both the 24 h and 72 h intervals. The control group had a lower percentage of CCL-13 cells in the G0/G1 phase compared with the SMG group at both time points, and it exhibited a higher total percentage of cells in the S and G2/M phases. The control group exhibited elevated levels of cell-cycle-related proteins, including cyclin A, cyclin D, and cdk6, compared with the SMG group. The flow cytometry results revealed that the apoptotic rate in the control group was significantly lower than that in the SMG group at both the 24 h and 72 h time points. However, the apoptotic percentage in the SMG group at the 72-h mark was significantly lower than that at the 24-h mark. SMG reduces the viability and proliferation ability of CCL-13 cells. After a period of recovery and adaptation to normal gravity conditions (1G), the CCL-13 cells in the SMG group showed better signs of recovery after 72 h than after 24 h.

## 1. Introduction

Living organisms are constantly subjected to gravitational force. Transitioning from the Earth’s gravitational conditions to a state of microgravity impacts the functioning of various systems, including the nervous and immune systems, and the bones, skin, and liver [1,2,3,4]. Assessing the impact of microgravity on the human body at the International Space Station (ISS) is challenging, leading to numerous experiments using devices that simulate these conditions, such as two-dimensional (2D) clinostats [5], three-dimensional (3D) clinostats [6], and random positioning machines (RPMs) [7].

Cell proliferation is crucial for survival and development. Simulated microgravity (SMG) can influence cell proliferation in diverse ways, varying with the cell type, culture conditions, and microgravity simulation apparatus used. The liver is a very important organ of essential metabolic processes since it interacts with almost every tissue in the human body; however, it is also a very sensitive organ. This has been reported in many studies, showing that rodent livers are sensitive to gravity conditions during spaceflight and even in ground-based experiments performed through simulation devices [4,8]. Hepatocytes make up approximately 80% of liver volume [9] and the commonly used hepatocyte model is Chang liver cells, which function similarly to normal hepatocytes [10]. Recent research indicates that SMG conditions may disrupt mitochondrial function, negatively impacting cell proliferation [11]. Studies on the impact of SMG on lung cancer cells have demonstrated that H1703 cells exhibit decreased proliferation when exposed to SMG [12]. Microgravity conditions also have adverse effects on immune cells. Studies conducted by Leonardo K. Martinelli and his team demonstrated that microgravity can lead to the reduced proliferation of immune cells. Their research also indicated that this effect intensifies over time, with a more pronounced decrease in immune cell proliferation observed at 48 h compared with 24 h [13]. In addition, microgravity conditions impact stem cells. Studies have indicated that microgravity can reduce the differentiation and regenerative capabilities of embryonic stem cells [14]. Human hematopoietic progenitor cells exhibited decreased proliferation under simulated microgravity conditions produced by rotating wall vessels (RWVs) [15]. It has been reported that the differentiation capacity of human bone marrow mesenchymal stem cells is also reduced under SMG conditions [16]. In addition, our team has explored the impact of SMG on various cell types. Research on porcine granulosa cells has shown that SMG conditions can reduce cell proliferation and change cell morphology [17]. Cell morphological alterations have been observed in 3T3 cells under simulated microgravity (SMG) [18]. Some research on Chang liver cells under SMG has also show decreased proliferation after 72 h of culture via the attenuation of the major cell cycle regulators and cytoskeletal proteins [19]. However, the recovery of the in vitro proliferation ability of CCL-13 cells after microgravity induction has not been well characterized. In particular, the changes in cell viability, cell cycle, and the expression of some proteins involved in cell cycle control have not been clarified. Therefore, this study was designed to evaluate the changes in cell viability, cell cycle, and some key cell cycle control proteins.

## 2. Materials and Methods

### 2.1. Cell Culture and SMG Induction

CCL-13 cells were cultured in 96-well plates (161093, Thermo Scientific, Waltham, MA, USA) at a density of 5 × 10^2^ cells/well and in T-25 Flasks (160430, Thermo Scientific, Waltham, MA, USA) at 5 × 10^4^ cells/flask in a CO_2_ incubator (MCO-18AIC, Sanyo Electric Co., Moriguchi, Osaka, Japan). DMEM/Ham’s F-12 (DMEM-12-A, Capricorn Scientific, Ebsdorfergrund, Germany) was used as the culture medium, supplemented with 10% FBS (FBS-HI-22B, Capricorn Scientific, Ebsdorfergrund, Germany) and 1% pen/strep (15140-122, Gibco, Waltham, MA, USA). All culture plates and flasks were filled with this medium and sealed with parafilm to prevent spillage during SMG testing. The cells were split into two groups: the simulated microgravity (SMG) group and the control group. The SMG group’s cells were cultured in a Gravite^®^ gravity controller (AS ONE INTERNATIONAL, INC., Santa Clara, CA, USA). The Gravite features four programs for simulating microgravity: mode A (×4 rpm), mode B (×3 rpm), mode C (×2 rpm), and mode D (×1 rpm). Mode C was suggested for the cell culture. The control group (1 G treatment) cell plates were put on the lower tray, while the SMG group’s cell plates were put in the Gravite on the upper tray of the same CO_2_ incubator. The CCL-13 from both groups were cultured at 37 °C and 5% CO_2_ for 72 h.

### 2.2. Recovery After SMG Induction

After the SMG induction test ended, the SMG and control groups were cultured under normal gravity conditions (1G) in a CO_2_ incubator at 37°C and 5% CO_2_. These two groups of experiments were conducted to evaluate the differences after 24 and 72 h.

### 2.3. WST-1 Assay

The WST-1 assay was used to evaluate CCL-13 cell proliferation. These cells were seeded at a density of 5 × 10^2^ cells/well in 96-well plates, and experiments were performed. Afterward, the stale culture medium in each well was replaced with 100 µL of a fresh medium and 10 µL of a WST-1 solution (11644807001, Roche, Basel, Switzerland). The 96-well plates were covered with foil to protect them from light and incubated at 37 °C and 5% CO_2_ for 3 h. Then, the plates were measured using the GloMax^®^ Explorer Multimode Microplate Reader (Promega, Madison, WI, USA) at an optical density of 450 (OD 450).

### 2.4. Cell Cycle Analysis

The 96-well plates were also used in the cell cycle analysis. The stale medium in the cell culture wells was removed, and the cells were washed with a phosphate-buffered saline (PBS 1X) (Gibco, Waltham, MA, USA) solution. The nuclei were stained with a Hoechst 33342 solution (14533, Sigma-Aldrich, Burlington, MA, USA) after 30 min and washed three times with PBS 1× (Gibco, Waltham, MA, USA). After that, the 96-well plates underwent cell cycle evaluation using the Cell Cycle App of the fluorescent Cytell microscope (GE Healthcare, Chicago, ILL, USA).

### 2.5. Flow Cytometry Analysis

The CCL-13 cells were seeded in T-25 flasks at a density of 5 ×10^4^ cells/flask and subjected to simulated microgravity. Subsequently, the culture medium was removed and substituted with the buffer from the FITC Annexin V Apoptosis Detection Kit I (556547, BD Biosciences, Franklin Lakes, NJ, USA). Cell cycle progression, FSC values, and cell viability were analyzed using a BD Accuri C6 Plus cytometer (BD Biosciences, Franklin Lakes, NJ, USA).

### 2.6. Western Blot Analysis

The CCL-13 cells harvested from the T-25 flasks were lysed using Optiblot LDS Sample Buffer (ab119196, Abcam, Waltham, MA, USA) in a thermostatic bath at 70 °C for 10 min. The lysate proteins were then loaded equally into a 4–12% SDS-PAGE gel and run with Optiblot SDS Running Buffer (ab119197, Abcam, Waltham, MA, USA) for 2 h at 50 V. Following electrophoresis, the proteins were transferred from the gel to a PVDF membrane (ab133411, Abcam, Waltham, MA, USA) for 2 h at 50 V. The membrane was blocked with a blocking buffer (ab126587, Abcam, Waltham, MA, USA) for 1 h at room temperature and then incubated with primary antibodies overnight at 4 °C. The primary antibodies used were anti-cyclin A1 + cyclin A2 (ab185619, Abcam, Waltham, MA, USA) and anti-cdk6 (ab124821, Abcam, Waltham, MA, USA) at a 1:5000 dilution, with anti-gapdh (ab181602, Abcam, Waltham, MA, USA) as the control at a 1:10,000 dilution. After incubation with the primary antibodies, the membrane was washed thrice with TBST for 10 min each time. Subsequently, they were incubated with goat anti-rabbit IgG HRP-conjugated secondary antibodies (ab6721, Abcam, Waltham, MA, USA) for 1 h at room temperature. The ECL Kit (ab65623, Abcam, Waltham, MA, USA) was used for blot visualization, and imaging was performed using X-ray films.

### 2.7. Microfilament Staining

Paraformaldehyde (4%) was applied to fix cells for 30 min, then cells were permeabilized with 0.1% Triton X-100 (Merck, Darmstadt, Germany) overnight at 4 °C. The microfilament was stained with Phalloidin CruzFluor™ 488 Conjugate (sc-363791; Santa Cruz Biotechnology, Santa Cruz, CA, USA) for 1 h. The cells were washed 3 times with PBS (Gibco, Thermo Fisher Scientific, Inc., Waltham, MA, USA) for 10 min each time. The nuclei were stained with Hoechst 33342 (14533; Sigma-Aldrich, St. Louis, MO, USA) for 30 min. The cells were washed 3 times with PBS for 10 min each time. The changes in microfilament bundles of cells were evaluated under the Cytell microscope.

### 2.8. Statistical Analysis

All experiments were repeated at least three times. Statistical analyses were performed with Sigma Plot 11.0 (Systat Software Inc., San Jose, CA, USA). The data obtained were analyzed using “one-way ANOVA”. Where *p* < 0.05 was considered statistically significant.

## 3. Results

### 3.1. CCL-13 Cell Proliferation

Under microscope observation, the CCL-3 cells in the SMG group showed a higher cytoplasm expansion compared with cells in the control group (Figure 1A). The proliferative capacity was estimated using the WST-1 assay. The OD value of the CCL-13 cells in the control group was higher than that in the SMG group after 24 h (1.754 ± 0.0288 vs. 1.5247 ± 0.0584, respectively) and after 72 h (3.8757 ± 0.0552 vs. 3.6167 ± 0.0329, respectively) (Figure 1B and Appendix A). These results showed that CCL-13 cells continued to proliferate after finishing simulated microgravity induction, however the proliferation ratio of these cells was lower than cells from the control group that were exposed to normal gravity condition.

### 3.2. CCL-13 Cell Viability

Flow cytometry was used to evaluate the viability of the CCL-13 cells after simulated microgravity induction. The results showed significant differences in CCL-13 cell viability after 24 h of recovery between the control and SMG groups (Figure 2 and Appendix A). Specifically, the control group had a significantly higher percentage of viable cells. This difference was statistically significant compared with the SMG group (97.75 ± 0.15 vs. 91.43 ± 0.60, respectively). The percentage of cells that survived after 72 h of recovery showed a similar trend, wherein the CCL-13 cells in the control group had a higher survival rate than those in the SMG group (97.53 ± 0.402 vs. 96.65 ± 0.155, respectively). This difference was statistically significant; however, the difference between the two experimental groups after 72 h of recovery was significantly less than that after 24 h. The images of the cell nuclei also show that cells in the SMG group (Figure 3B,D) had more fragmentation than those in the control group (Figure 3A,C) after 24 h and 72 h of recovery.

The actin staining results show changes in the actin filament structures in the apoptotic cells in the SMG group. The actin structure changed to different levels depending on the stage of apoptosis (Figure 4). In addition, the Western blot experiment showed similar results, where the BAX protein’s expression level in the control group was lower than that in the SMG group (Figure 3E). These results indicate that CCL-13 cells after experience microgravity induction exhibited some characteristics of apoptosis more strongly than the control group. The strong expression of apoptosis could potentially be a factor in the SMG group’s CCL-13 cells’ noticeably reduced cell viability compared with the control group.

### 3.3. Cell Cycle Evaluation

The cell cycle of CCL-13 cells was evaluated using a Cytell electron microscope. The CCL-13 cells in the control group showed statistically significant differences at 24 h and 72 h compared with the SMG group (Figure 5A,B and Appendix A). Specifically, the control group had a G0/G1 cell ratio of 51.60 ± 1.9758 at 24 h, lower than the SMG group’s G0/G1 cell ratio of 54.79 ± 1.8643. Results with a similar trend were obtained at the 72 h experimental mark, where the ratio of cells in the G0/G1 phase in the control group was lower than that in the SMG group (49.8 ± 0.9183 and 52.95 ± 1.1853, respectively). Conversely, the total ratios of CCL-13 cells in the S and G2/M phases at 24 and 72 h in the control group (38.89 ± 2.52 and 44.06 ± 0.85, respectively) were higher than those in the SMG group (33.51 ± 0.85 and 41.85 ± 0.91, respectively). These results reveal that SMG can increase the ratio of CCL-13 cells in the G0/G1 phase, which causes CCL-13 cells to enter the cell cycle arrest phase while reducing the proportion of cells entering the proliferative phase by reducing the number of cells entering the S and G2/M phases.

### 3.4. The Expression of Cell-Cycle-Related Proteins

The expression of cell-cycle-related proteins was assessed using the Western blot method (Figure 6). The experimental results showed lower expression levels of cell-cycle-related proteins including cdk6, cyclin A, and cyclin D in the SMG group than in the control group at the 24 h (Figure 6A) and 72 h time points (Figure 6B). This downregulation of cell-cycle-related proteins resulted in an induction of the arrest phase in CCL13 cells from the SMG group.

## 4. Discussion

Prior research indicates that simulated microgravity (SMG) conditions can decrease cell proliferation capabilities [17,18,19]. In this investigation, the WST-1 assay outcomes revealed a reduced proliferation rate of CCL-13 cells under SMG conditions. This reduced proliferation may arise due to various factors, often cell cycle alterations. The effect of SMG on the CCL-13 cell cycle observed in this study included an increased number of cells entering the G0/G1 phase and a reduced cell count progressing to the G2/M and S phases, leading to decreased cell division and proliferation in the SMG group. These findings align with other research, such as Li et al.’s work, which demonstrated a similar increase in cells entering the G0/G1 phase when human bone marrow mesenchymal stem cells were cultured under SMG conditions [16]. Deng et al. reported that U251 glioma cells experienced G2/M-phase arrest under SMG conditions [20].

Cell cycle phase changes are closely related to the expression of cyclins and cdks [21]. The function of cdk6 in cell cycle progression, and viability requires binding to D-type cyclin to enable kinase-dependent protein phosphorylation [22]. Cyclin D plays a central role in the regulation of proliferation, linking the extracellular signaling environment to cell cycle progression [23]. Experiments have observed that cyclin D levels are typically elevated during the G1 and G2 phases of the cell cycle [24]. Cyclin D binds to cdk4 and cdk6 to promote cell cycle progression from the G0/G1 phase to the S phase [25]. Cyclins A1 and A2 are crucial in the S phase and G2/M transitions [26,27]. During the S phase, cyclin A2 regulates the initiation and progression of DNA synthesis in the nucleus [28]. Cyclin A concentrations increase with the onset of the S phase [29]. In this study, the expressions of cdk6, cyclin D, and cyclin A generally declined in the group subjected to SMG, particularly at the 24 h mark. Although these protein levels rose in the SMG group at the 72 h interval, they remained notably lower than those in the control group. This observation correlates with the increased entry of CCL-13 cells into the G0/G1 phase and the reduced transition into the S and G2/M phases observed in the cell cycle analysis.

Apoptosis is another cause of reduced cell proliferation. Many studies have shown that exposure to microgravity conditions can increase apoptosis in benign and cancerous cells [30,31,32]. This phenomenon happens naturally when apoptosis is inherently a protection reaction and can be triggered by oxidative stress, noxious agents [33], or radiation [34], following the intrinsic apoptotic pathway. This apoptotic process is intricately linked to the BCL-2 family proteins. BAX (specifically, the BCL-2-associated X protein) was initially identified as a pro-apoptotic member of the BCL-2 protein family. BAX, which shares extensive amino acid similarity with BCL-2, is concentrated within the highly conserved I and II domains. Overexpression of BAX hastened apoptotic cell death induced by cytokine deprivation in an IL-3-dependent cell line in [35]. This research showed that the number of CCL-13 cells entering apoptosis in the SMG-exposed group was much higher than in the control group, especially at 24 h. This percentage was reduced in the SMG group at 72 h but remained higher than in the control group. The expression level of the BAX protein showed a similar trend, in which the expression level in the SMG group was significantly higher than in the control group at the 24 h and 72 h time points.

The cytoskeleton is a microgravity-sensitive structure. Regulation of the cytoskeletal network occurs after just seconds to hours of exposure to microgravity conditions [36,37]. The cytoskeleton includes microtubules, intermediate filaments, and microfilaments, of which microtubules and microfilaments are essential for cell division [38,39]. Microfilaments are made of the structural protein actin, which is often found in two forms: the polymer (F-actin) and monomer (G-actin) forms. Monomer actin, also known as spherical actin, is a spherical molecule composed of a polypeptide chain. Actin polymers are also known as fibroactin since they are responsible for producing actin filaments. The actin cytoskeleton is a crucial subcellular filament system [40]. It regulates fundamental processes, such as cell division, muscle contraction, cell mobility, and tissue integrity, by interacting with various proteins and regulatory cells [40]. Early cytoskeletal research on apoptotic cell death connected actin filament disintegration, rearrangement, and cleavage by caspases to late-stage alterations in cell adhesion and morphology [41,42]. Actin is drawn to the mitochondria at the beginning of apoptosis at the time of mitochondrial permeabilization [43,44]. In this study, the F-actin imaging results differed between the SMG and control group cells. The F-actin structure became unstable and formed actin-rich territory with the appearance of many apoptotic cells and cell nucleus fragmentation in the SMG group (Figure 4D,E). This result resembled the research by King et al., who reported that apoptosis could make F-actin-rich territory respond to DNA damage [45].

## 5. Conclusions

The present study demonstrated that SMG can induce a reduction of CCL-13 cell proliferation by altering the cell cycle and inducing cells to enter the apoptosis process. Moreover, this study also found that CCL-13 cells retrieved the proliferation at 24 h and 72 h after returning to 1G condition. However, their proliferation was lower than the control group. Interestingly, the higher proliferation and viability of CCL-13 cells were observed at 72 h compared with 24 h. The results of this study enhance our understanding of cell proliferation recovery after simulated microgravity (SMG) exposure. This process is accelerated when cells are cultured under normal gravity conditions (1G).

## Figures and Tables

**Figure 1 cimb-47-00164-f001:**
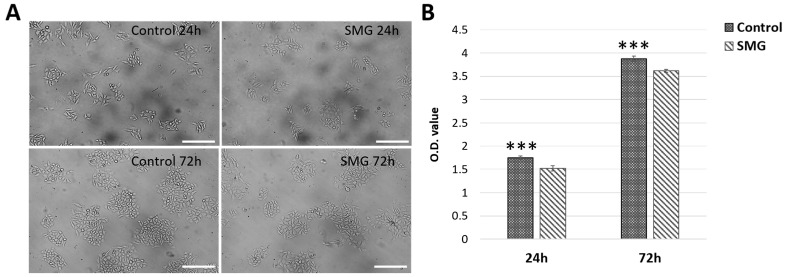
The sample preparation and cell proliferation. (**A**) Cell morphology; (**B**) the O.D. value was measured using WST-1 assay. *** indicates a significant difference between the control and SMG groups (*p* < 0.001). Scale bar = 223.64 µm.

**Figure 2 cimb-47-00164-f002:**
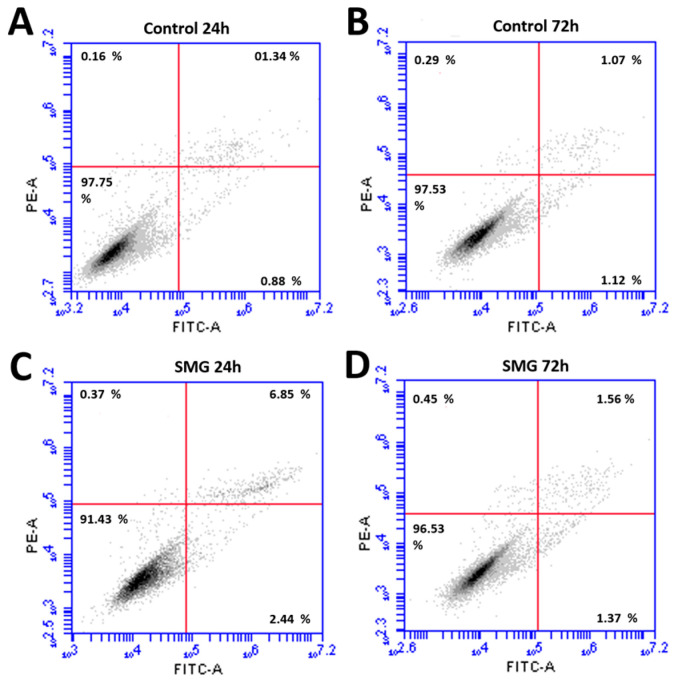
The viability and apoptosis of CCL-13 cells were assessed using the flow cytometry system. (**A**) Cells in the control group at 24 h; (**B**) cells in the control group at 72 h; (**C**) cells in the SMG group at 24 h; and (**D**) cells in the SMG group at 72 h.

**Figure 3 cimb-47-00164-f003:**
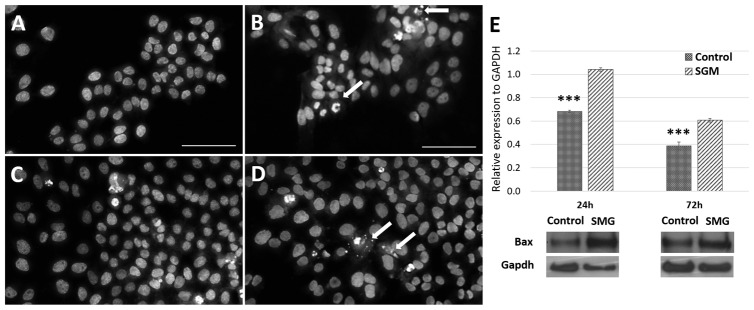
Images of fragmented CCL-13 nuclei during apoptosis and BAX expression. (**A**) Cells in the control group at 24 h; (**B**) cells in the SMG group at 24 h; (**C**) cells in the control group at 72 h; (**D**) cells in the SMG group at 72 h. (**E**). BAX expression was evaluated using Western blot. *** indicates a significant difference between the control and SMG groups (*p* < 0.001). Arrows indicate nuclear regions affected by apoptosis. Scale bar = 50 µm.

**Figure 4 cimb-47-00164-f004:**
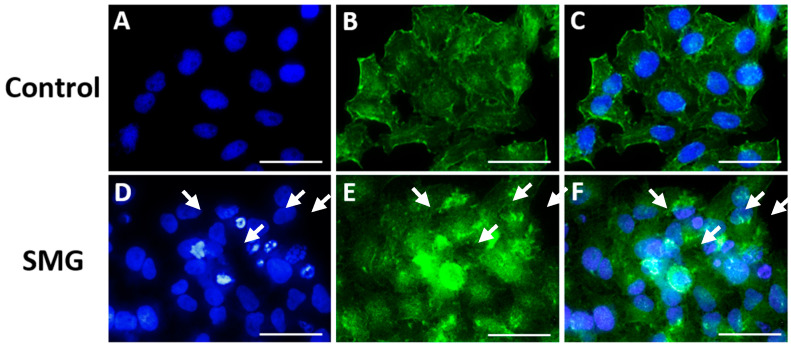
Nuclei and F-actin staining images of CCL-13 cells. (**A**) Nuclei staining in the control group; (**B**) F-actin staining in the control group; (**C**) merged image of F-actin and nuclei staining in the control group; (**D**) nuclei staining in the SMG group; (**E**) F-actin staining in the SMG group; and (**F**) merged image of F-actin and nuclei staining in the SMG group. Arrows indicate nuclear regions and F-actin structures affected by apoptosis. Scale bar = 30 µm.

**Figure 5 cimb-47-00164-f005:**
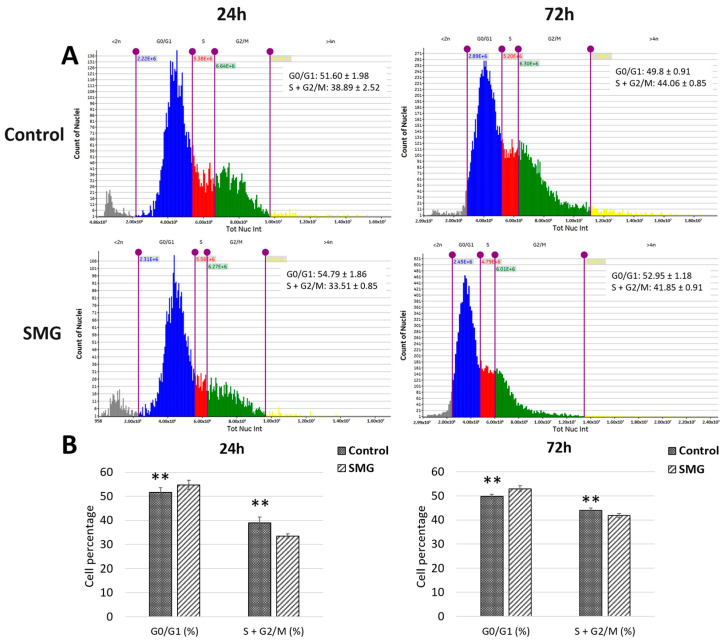
The CCL-13 cycle progression from 24 h to 72 h between the two groups. (**A**) The distribution of cell stages analyzed using the Cell Cycle App (Cytell microscope). The G0/G1 phase is represented in blue, the S phase in red, and the G2/M phase in green. (**B**) The ratio of cells in cell cycle phases. ** indicates a significant difference between the control and SMG groups (*p* < 0.01).

**Figure 6 cimb-47-00164-f006:**
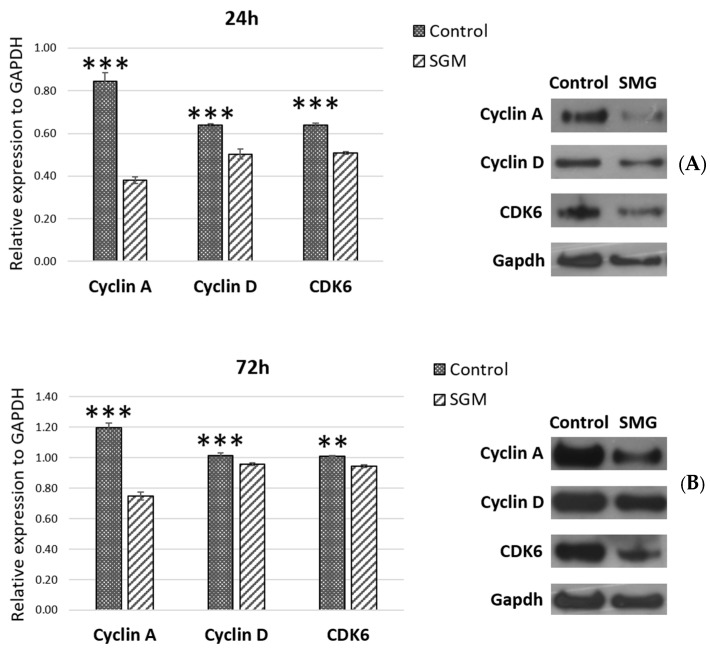
The expression levels of cell-cycle-related proteins in the CCL-13 cells in the control and SMG groups at 24 h (**A**) and 72 h (**B**). ** indicates a significant difference between the control and SMG groups (*p* < 0.01). *** indicates a significant difference between the control and SMG groups (*p* < 0.001).

## Data Availability

Data are contained within the article.

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
