# Peer review of "The Proliferation of Chang Liver Cells After Simulated Microgravity Induction"

_cimb, 2025, doi:10.3390/cimb47030164_

Round 1
Reviewer 1 Report
Comments and Suggestions for Authors
The manuscript entitled "The proliferation of Chang liver cells after simulated microgravity induction" investigates the regenerative capacity of CCL-13 cells after the induction of SMG.
The issue discussed in this work is very interesting, but requires several corrections.
- There are no attached supplementary materials, so it's hard for me to comment on them.
- It is required to review the entire text for errors such as line 16: fol-lowing; line 21 percent-age. Grammar correction is also required.
Author Response
Dear Prof. Dr. Madhav Bhatia, Editorial Board and Reviewers,
We are very grateful to the Editor for your consideration of our manuscript. We would like to thank the Reviewers for careful and thorough reading of manuscript and for the thoughtful comments and constructive suggestions, which help to improve the quality of this manuscript. Each comment has been carefully considered point by point and responded. Responses to the reviewers and changes in the revised manuscript are as follows.
Reviewer 1:
Comment 1. There are no attached supplementary materials, so it's hard for me to comment on them.
Response 1. Thank you for your comment. The supplementary data has been uploaded to the manuscript dashboard
Comment 2. It is required to review the entire text for errors such as line 16: fol-lowing; line 21 percent-age. Grammar correction is also required.
Response 2. These errors of the text have been corrected in the manuscript
Comment 3. In the Materials and Methods section 2.1 I would remove 'Figure 1A - let it stay only in the Results section.
Response 3. “Figure 1A” and “Figure 1B” have been removed from the text (line 77, 81, and 83).
Comment 4. In section 2.1 the authors suddenly use the abbreviation CT - please explain what it is.
Response 4. We have replaced "CT group" by "control group".
Comment 5. In the Results section 3.1 the authors write that few morphological changes were observed - please describe what these changes are.
Response 5. Thank you for your comment. The sentence “Few morphological changes between the two experimental groups were observed during the in vitro culture (Figure 1A).” was rewritten to “Under microscope observation, the CCL-13 cells in SMG group showed a higher expansion of cytoplasm compared to the cells in the control group (Figure 1A).” to clarify the change in CCL-13 cells under SMG condition. (line 147-148)
Comment 6. At the end of section 3.1, 3.2 and 3.4- one sentence summarizing the obtained results (what they mean).
Response 6. Thank you for your comment. A summary has been added to clarify the research results in sections 3.1, 3.2, and 3.4.
We hope that the revision of our manuscript could meet your requirement.
Best regards

Reviewer 2 Report
Comments and Suggestions for Authors
The introduction provides good context on microgravity effects, but lacks a clear hypothesis statement. The authors should explicitly state their research hypothesis before describing the study aims.
Some of the literature review in introduction appears fragmented. Consider reorganizing to present a more logical flow from general microgravity effects to specific impacts on liver cells.
p-values throughout the manuscript should be consistent.
Line 82: The specific model/parameters of the Gravite® gravity controller should be provided
Line 86: The exact conditions for the "normal gravity" control group should be specified
Line 91: The cell counting method before WST-1 assay is not described.
Figure 1: Scale bars are missing in the cell morphology images
The discussion effectively interprets the results but could be strengthened by adding comparison with other liver cell types under microgravity.
Author Response
Dear Prof. Dr. Madhav Bhatia, Editorial Board and Reviewers,
We are very grateful to the editor for your consideration of our manuscript. We would like to thank the reviewers for their careful and thorough reading of the manuscript and for the thoughtful comments and constructive suggestions, which help to improve the quality of this manuscript. Each comment has been carefully considered point by point and responded. Responses to the reviewers and changes in the revised manuscript are as follows:
Reviewer 2
Comment 1. The introduction provides good context on microgravity effects but lacks a clear hypothesis statement. The authors should explicitly state their research hypothesis before describing the study aims. Some of the literature review in the introduction appears fragmented. Consider reorganizing to present a more logical flow from general microgravity effects to specific impacts on liver cells.
Response 1. Thank you so much for your comment. We have added extra information in the introduction of this manuscript to present the reason why we carry out this study, as followed from line 69 to 74
Comment 2. p-values throughout the manuscript should be consistent.
Response 2. In this study, we used the asterisk to indicate a significant difference between the control and SMG groups. We have corrected the P value in Statistical Analysis of Materials and Methods.
Comment 3. Line 82: The specific model/parameters of the Gravite® gravity controller should be provided
Response 3. The specific model/parameters of the Gravite® gravity controller have been provided in the Materials methods, as follows: “The Gravite features four programs for simulating microgravity: mode A (×4 rpm), mode B (×3 rpm), mode C (×2 rpm), and mode D (×1 rpm). The mode C was suggested for cell culture.” (line 86-88)
Comment 4. Line 86: The exact conditions for the "normal gravity" control group should be specified
Response 4. The "normal gravity" is understood as Earth gravity condition (1G ). We have added the information about the sample position setting in CO2 incubator as followed: “The control group (1G treatment) cell plates were put on the lower tray, while the SMG group's cell plates were put in Gravite on the upper tray of the same CO2 incubator. The CCL-13 from both groups were cultured at 37 °C, 5% CO2 for 72 h.” (line 88-91)
Comment 5. Line 91: The cell counting method before WST-1 assay is not described.
Response 5. In the Meterials and Methods, we have presented the seeding cell number for WST-1 assay at a density of 5 × 102 cells/well in 96-well plates (line 98). All wells were designed for WST-1 assay. In the further research, we will carry out an experiment of cell counting and the WST-1 assay together for the determination of cell proliferation.
Comment 6. Figure 1: Scale bars are missing in the cell morphology images
Response 6. The Scale bars has been added to this figure 1A
Comment 7. The discussion effectively interprets the results but could be strengthened by adding comparison with other liver cell types under microgravity.
Response 7. At this time, we could not find information regarding the recovery of proliferative ability after microgravity induction of other liver cells. Therefore, we are currently unable to make further comparisons. We hope that in the further research, we could approach more information about changes in liver cell proliferation post-simulated microgravity induction to clarify the alterations in structure and function of these cells.
We hope that the revision of our manuscript could meet your requirement.
Best regards